# IF-Garments: Reconstructing Your Intersection-Free Multi-Layered Garments from Monocular Videos

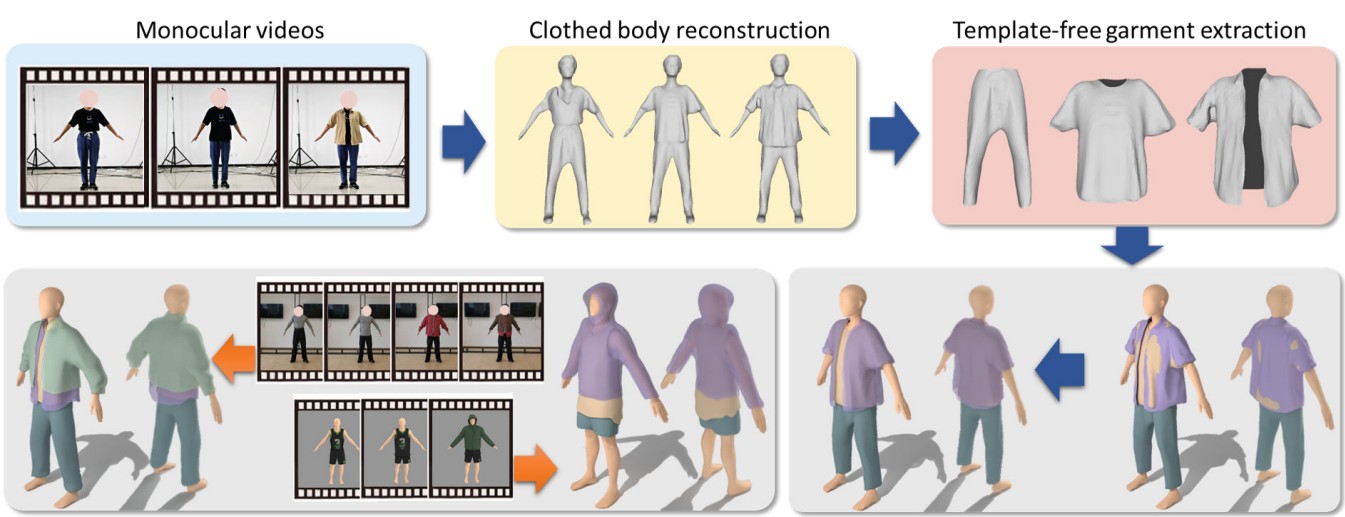

**Figure 1: We propose IF-Garments to reconstruct Intersection-Free Garments from monocular videos. The template-free garment extraction shows great generalization while the physics-aware correction robustly eliminates inter-layer penetration.**

## ABSTRACT

Reconstructing garments from monocular videos has attracted considerable attention as it provides a convenient and low-cost solution for clothing digitization. In reality, people wear clothing with countless variations and multiple layers. Existing studies attempt to extract garments from a single video. They either behave poorly in generalization due to reliance on limited clothing templates or struggle to handle the intersections of multi-layered clothing leading to the lack of physical plausibility. Besides, there are inevitable and undetectable overlaps for a single video that hinder researchers from modeling complete and intersection-free multi-layered clothing. To address the above limitations, in this paper, we propose a novel method to reconstruct multi-layered clothing from multiple monocular videos sequentially, which surpasses existing work in generalization and robustness against penetration. For each video, neural fields are employed to implicitly represent the clothed body, from which the meshes with frame-consistent structures are explicitly extracted. Next, we implement a template-free method for

extracting a single garment by back-projecting the image segmentation labels of different frames onto these meshes. In this way, multiple garments can be obtained from these monocular videos and then aligned to form the whole outfit. However, intersection always occurs due to overlapping deformation in the real world and perceptual errors for monocular videos. To this end, we innovatively introduce a physics-aware module that combines neural fields with a position-based simulation framework to fine-tune the penetrating vertices of garments, ensuring robustly intersection-free. Additionally, we collect a mini dataset with fashionable garments to evaluate the quality of clothing reconstruction comprehensively. The code and data will be open-sourced if this work is accepted.

## CCS CONCEPTS

• **Information systems** → **Multimedia content creation**; • **Computing methodologies** → **Shape modeling**.

## KEYWORDS

Clothing reconstruction, Clothing simulation

## 1 INTRODUCTION

Digitalization of clothing holds significant importance in livestream sales, virtual try-ons, and entertainment. Recent work[18, 24, 45] has shown progress in extracting garments from monocular images or videos, which becomes highly convenient and accessible for general commerce. In daily life, people wear multiple layers of garments varying in different styles, which indeed poses significant challenges for high-quality reconstruction.

We start the discussion by reconstructing a single garment with a monocular image or video, which can be categorized into template-based and template-free approaches. Template-based approaches[12, 15, 24, 29, 45, 48, 51] choose and deform a pre-designed garment template to fit observations. They suffer from a limited set of templates, lacking generalization. Template-free methods[17, 18, 53] first reconstruct the 3D clothed body, then separate the clothing from the body using 2D segmentation of garments. To obtain the clothed body, one straightforward idea is to use scanning devices[31, 53, 56], which needs costly manual post-processing. With the parametric human body models[33, 43], some studies[4, 11, 17, 18, 23] consider clothing as the offset of body vertices, which is feasible for most garments as they usually adhere to the human body in rest pose. For multi-layered outfits, existing work aims to extract multiple pieces of clothing from a single monocular image or video simultaneously. They either treat all clothing as a single entity[17, 18] or support only two pieces of clothing[24, 29, 45, 51]. Those[2, 15, 39] attempting to overcome three or more layers of clothing suffer from distortions caused by overlap regarding inner garments. Besides, they suffer from occlusion in a single video so the inner layer of outfits is incomplete or in the wrong type.

Due to the inevitable occlusion, it is an ill-posed problem to recover multi-layered clothing from a single video, but is possible to extract an uncovered garment. Further, multiple garments can be obtained from corresponding videos and then aligned to form a multi-layered outfit. Therefore, we propose a novel methodology to sequentially reconstruct each layer of clothing from multiple monocular videos. As shown in Figure 2, we ask the actor to remove occlusion for the target garment in each video and rotate slowly to provide approximate multi-view information Our full approach is divided into three parts. First, a neural Signed Distance Field (SDF)[23, 42, 50] is leveraged to represent the clothed body and extract the meshes with consistent topology via marching cube[34]. Second, a template-free approach is implemented by back-projecting image segmentation labels of the garment from different frames onto these meshes' vertices. The labeled vertices are then gathered to form the garment, resulting in multiple garments from corresponding videos. Finally, it is crucial to combine these garments into a multi-layered outfit. However, it indeed introduces intersections due to the following reasons: i) in the real world, the movement and overlap of clothing can cause deformation; ii) for reconstructing, results obtained from different videos inevitably contain misaligned errors due to monocular depth ambiguity. To robustly eliminate inter-layer penetration, a physics-aware module is proposed with a novel pipeline to fine-tune garments from the outer to the inner. Concretely, given an SDF of the clothed body of the inner garment layer, we query the signed distance of the outer layer's vertices and push out those vertices with negative signs along the intersecting direction. To further ensure physically plausible deformation, the SDF-based penetration handling is implemented in a position-based simulation framework[36] with carefully devised physics constraints. In addition, existing datasets[4, 18, 23] have limited clothing variety, making it difficult to adequately evaluate generalization and also struggle to meet the requirements for multi-layered clothing reconstruction. Thus, we create *mini-IFG*, a small dataset with 23 videos collected from both physics simulation and the real world for comprehensive evaluation.

**Table 1: Comparison among ours and existing works. Our method supports multi-layer clothing reconstruction without intersection and is independent of garment templates.**

| Research | Layers | Template-free | Intersection-free |
|---|---|---|---|
| PERGAMO[12] | 1 | ✗ | - |
| SCARF[18] | 1 | ✓ | - |
| DELTA[17] | 1 | ✓ | - |
| REC-MV[45] | 2 | ✗ | ✗ |
| BCNet[24] | 2 | ✗ | ✗ |
| Li *et al.*[29] | 2 | ✗ | ✗ |
| MulayCap[51] | 2 | ✗ | ✓ |
| SMPLict[15] | ≥3 | ✗ | ✗ |
| ClothWild[39] | ≥3 | ✗ | ✗ |
| LGN[2] | ≥3 | ✗ | ✓ |
| Ours | ≥3 | ✓ | ✓ |

Briefly, this paper represents the first attempt to reconstruct intersection-free and complete multi-layered clothing from several monocular videos. In Table 1, we make a comparison of studies related to clothing reconstruction from monocular images or videos. Our method surpasses existing work in terms of the number of clothing layers, generalization, and robustness against penetration. Experiments demonstrate that our method can confidently reconstruct challenging garments and robustly eliminate intersections.

The contributions of this paper can be summarized as follows:

- A template-free approach with great generalization for extracting a complete garment.
- A physics-aware module ensuring multi-layered clothing intersection-free with excellent robustness and quality.
- A small dataset containing 23 self-rotating videos of actors wearing fashionable garments.

## 2 RELATED WORK

We briefly review work related to recovering clothing from images or videos from the following three aspects.

*Clothed Body Reconstruction.* Some statistical human body models have been proposed by fitting a function with shape and pose to real 3D scans[5, 33, 41, 43]. It has made great progress [26, 27, 30, 52, 57, 58] in predicting body parameters from images and videos, which lays the foundation for clothed body reconstruction. PIFu[46] and PIFuHD[47] have pioneered the extraction of pixel-aligned spatial features from images and mapping them to implicit fields to reconstruct people in arbitrary poses and clothing. Follow-up methods[19–21, 54, 55, 59] then introduce parametric models as 3D features to condition implicit fields. These methods all require expensive 3D scans for supervision. Unlike single images, videos contain richer perspectives. Some studies[3, 4, 16, 23] treat clothing as offsets of body vertices. They establish a clothed body in canonical space based on parameterized models[33, 43], then learn mappings from the canonical space to the posed space of video frames to recover clothing. In self-rotating videos, clothing has minimal deformation. Thus, we follow SelfRecon[23] to learn neural fields from self-rotating videos to obtain the clothed body.

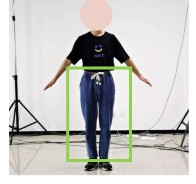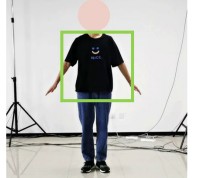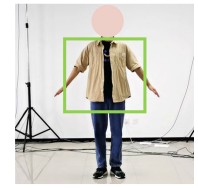

Layer1 (Pants)        Layer2 (T-shirt)        Layer3 (Shirt)

**Figure 2: Videos with the corresponding uncovered garment. The target garment is marked with a green rectangle.**

*Garments Extraction.* Researchers have already attempted to construct various clothing templates[8, 11, 24, 39, 51, 60]. Subsequent works fine-tune these templates to match observations. MGN[11] utilizes a large-scale clothing dataset to develop a per-category parametric model. BCNet[24] generates a rough template, which is subsequently enhanced with surface details through a displacement network. REC-MV[45] simultaneously optimizes the explicit feature curves and the implicit field of the garments, resulting in superior dynamic garment surfaces. However, the limited templates are insufficient to cover the vast diversity of clothing. Template-free methods extract garments from the clothed body, which is achieved by inverse mapping 2D segmentation to the 3D space. SCARF[18] represents the clothing using NeRF[38], which results in high-quality rendering but low-quality geometry. Similar to our template-free method, Xiang *et al.*[53] use 140 synchronized cameras to extract high-fidelity clothing. In contrast, our back-projection scheme leverages the geometric consistency of meshes across different frames, thus requiring only one camera.

*Multi-Layered Clothing Reconstruction.* Some studies learn a generative clothing model with neural distance field[13, 37, 42] from a 3D clothing dataset[15, 39]. They can decode each garment from the latent space and align them but lack consideration for penetration. LGN[2] leverages SDF[42] to propose a garment indication field to handle penetration but overlooks clothing deformation. In comparison, we implement a physics-aware module that combines SDF and physics constraints to eliminate inter-layer penetration while preserving natural non-rigid deformations.

## 3  METHODOLOGY

In Figure 3, we present an overview of our method, IF-Garments, which aims to reconstruct intersection-free multi-layered clothing from monocular videos faithfully. Our key insight lies in resourcefully leveraging the SDF's geometric and physical characteristics, employed in clothing extraction and inter-layer penetration correction respectively. Specifically, we first follow SelfRecon[23] to learn a neural SDF in canonical space to reconstruct the clothed body, which can be mapped to posed space by a pose-conditioned deformation field (Section 3.1). Next, we back-project the segmentation image labels from different viewpoints onto the vertices of the corresponding posed mesh to extract garments (Section 3.2). Finally, all of these garments are aligned in canonical space, and the physics-aware module solves the intersections between them (Section 3.3).

## 3.1  Clothed Body Reconstruction

Given a self-rotating video with $N$ frames, we adopt VideoAvatar[4] to generate the camera intrinsic $\pi$, and SMPL[33] parameters of the initial shape $\beta$, and per-frame's pose $\{\theta_i | i \in 1, \ldots, N\}$ and translation $\{t_i | i \in 1, \ldots, N\}$.

*Canonical Representation.* The clothed body $\mathcal{S}_\eta$ in canonical space with an A-pose is represented as the zero-level-set of an neural SDF[42], which is parametrized by a Multi-Layer Perceptron (MLP) $\phi$ with learnable weights $\eta$:

$$\mathcal{S}_\eta = \{\mathbf{x} \in \mathbb{R}^3 | \phi(\mathbf{x}; \eta) = 0\}, \qquad (1)$$

where the mesh of the clothed body $\mathcal{M}$ is extracted by marching cube[34].

*Deformation Field.* To map the clothed body in canonical space to posed space to match supervision from the video, we decompose the deformation field $\mathcal{D}$ into skinning transformation $\mathcal{W}$ and non-rigid deformation $d$. $\mathcal{W}$ ensures that the garment's surface deforms with the body's large-scale motions[20], which takes $\theta_i$ as the parameter and is pre-computed as described in [23]. An MLP $d$ with learnable weights $\psi$ models the fine-grained changes. In $i$-th frame, $d$ is conditioned by an optimizable $\mathbf{h}_i$ variable to apply deformations to points in the canonical space. By compositing $\mathcal{W}$ and $d$, we get the final deformation field $\mathcal{D} = \mathcal{W}(d(\cdot))$. It takes $\mathbf{h}_i$ and $\theta_i$ as input and transforms canonical points to the $i$-th frame posed space. We train $\mathcal{S}$ and $\mathcal{D}$ in the same manner as SelfRecon[23].

For brevity of description, we use $\mathcal{D}_i$ to denote $i$-th frame's deformation field, $\mathcal{S}_i$ for $i$-th frame's zero-level-set $\mathcal{D}_i(\mathcal{S}_\eta)$, and $\mathcal{M}_i$ for the mesh extracted from $\mathcal{S}_i$ via marching cube[34].

## 3.2  Template-Free Garment Extraction

In monocular self-rotating videos, the actor rotates slowly, providing approximate multi-view information about the clothing. Thanks to the deformation field discussed in Section 3.1, we can obtain clothed body mesh $\mathcal{M}_i$ in posed space. Especially, $\mathcal{M}$ and $\mathcal{M}_i$ in all frames share the same topology. Inspired by this, we propose a template-free clothing extraction method by back-projecting the garment's segmentation labels of video frames onto corresponding $\mathcal{M}_i$. These labels are shared across all of the canonical and posed meshes and aggregate to form the clothing $\mathcal{G}$ (see Figure 3(a)).

*Back-Projection.* Back-projection is realized by projecting 3D vertices forward onto 2D pixels to assign garment labels to $\mathcal{M}_i$. The parameters estimated from the video regarding $\pi$, $\theta_i$, and $t_i$ are referenced in the camera coordinate[4]. For $i$-th frame, we obtain $\mathcal{M}_i'$ by applying the deformation field and translation to $\mathcal{M}$:

$$\mathcal{M}_i' = \mathcal{D}_i(\mathcal{S}_\eta) + t_i = \mathcal{M}_i + t_i. \qquad (2)$$

Given a vertex $P = [X, Y, Z]^T \in \mathcal{M}_i'$, we render it onto the image plane as $p = [u, v]$ by the perspective projection:

$$\lambda \begin{bmatrix} u \\ v \\ 1 \end{bmatrix} = KP = \begin{bmatrix} \alpha_u & 0 & u_0 \\ 0 & \alpha_v & v_0 \\ 0 & 0 & 1 \end{bmatrix} \begin{bmatrix} X \\ Y \\ Z \end{bmatrix}, \qquad (3)$$

where $\lambda$ is the depth factor and equal to $Z$, $[u, v]$ are the 2D position in the image, $[u_0, v_0]$ and $[\alpha_u, \alpha_v]$ are the center and focal length of the camera intrinsic $\pi$ respectively. However, Equation (3)

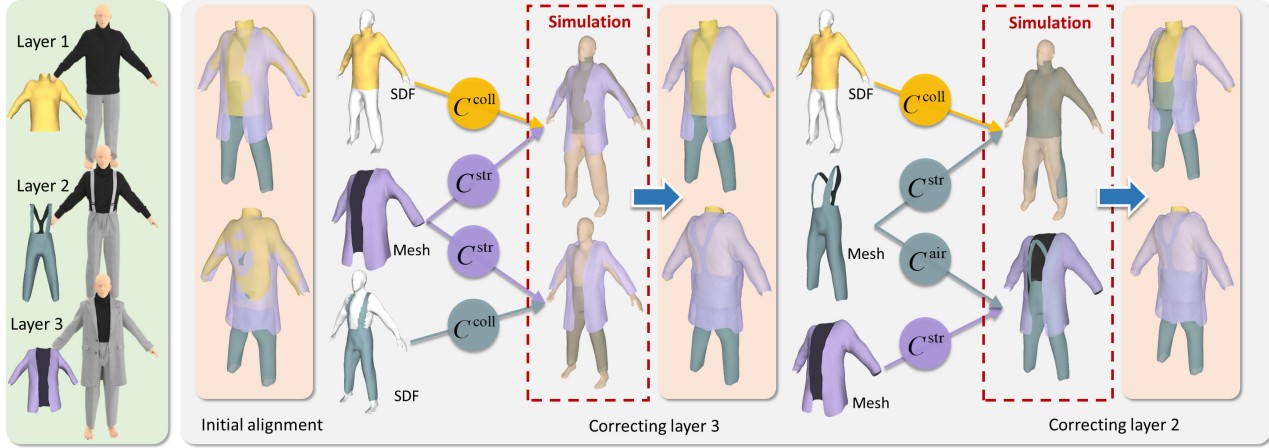

(a) Extracting an unoccluded garment from a monocular video.

(b) The physics-aware module to obtain intersection-free clothing.

**Figure 3: Overview of IF-Garments. (a) Following SelfRecon[23], we employ neural fields to reconstruct the clothed body from a monocular video, which can be mapped into the posed space by the deformation field. The garment is extracted by back-projecting the segmentation labels to the posed clothed body's vertices from multiple viewpoints. (b) The physics-aware module involves a novel pipeline to eliminate penetration from outer to inner in canonical space, accomplished with carefully designed constraints of $C^{\text{str}}$, $C^{\text{coll}}$, and $C^{\text{air}}$.**

ignores the visibility of vertices, leading to multiple vertices being projected onto the same pixel. To address this issue, we utilize the z-buffer algorithm from OpenGL[49], where the visibility of vertices is determined based on $\lambda$. Then, visible vertices will acquire the corresponding label on the segmentation image. Finally, $\mathcal{G}$ is composed of these labeled vertices. To mitigate the noise caused by segmentation errors, post-processing is employed, including isolated elements removal, hole filling, and Laplacian smoothing[14]. Depending on the complexity of the garment, we typically back-project 4 to 8 frames and fuse their labels.

## 3.3 Physics-aware Module

The clothed body in canonical space undergoes no rigid transformation, where all garments are aligned initially. However, penetration is inevitable due to the following reasons: i) the way clothing is worn varies slightly in each video to eliminate occlusion, and each additional garment will cause deformation to existing ones; b) perceptual errors arise due to the depth ambiguity of monocular videos. It poses a significant challenge for solving such penetration due to the lack of ground truth. Inspired by collision

detection[6, 22, 35] in computer graphics, the neural SDF is employed to handle intersections. To ensure physical plausibility, we conscientiously design 3 physics constraints in a position-based simulation framework[7, 36, 40]. Due to the undetectable overlap, it is impossible to model multi-layered clothing that completely adheres to the real world. However, we strive to address severe penetration through deformation as physically as possible.

### 3.3.1 Simulation Framework.
Following [36], we establish physics constraints between vertices and solve them with the Gauss-Seidel method. In each time step, the positions of the vertices are projected onto each constraint manifold along the constraint gradient. Due to space limitations, the solving pipeline can be found in the supplementary materials. Here, we primarily discuss the physics constraints devised in our work.

Given a vertex $\mathbf{p}$ and the vertex $\bar{\mathbf{p}}$ forming an edge with $\mathbf{p}$, we define the stretch constraint as:

$$C^{\text{str}}(\mathbf{p}, \bar{\mathbf{p}}) = \| \mathbf{p} - \bar{\mathbf{p}} \| - L, \tag{4}$$

where $\| \cdot \|$ means Euclidean norm, and $L$ is the rest length. For a garment mesh $\mathcal{G}$, $C^{\text{str}}$ accounts for the non-rigid deformation and is applied to all vertices.

*3.3.2 SDF-Based Collision Detection.* We query the penetration status of a point in the SDF, including the signed distance (penetration depth) and the gradient (penetration direction), which provides robustness for handling intersections. Given a query point $\mathbf{p}$ and an SDF $\phi$, we define the collision constraint as:

$$C^{\text{coll}}(\mathbf{p}) = \phi(\mathbf{p}) - \alpha \geq 0, \tag{5}$$

where $\alpha$ is a small positive value to enhance robustness. When $C^{\text{coll}}(\mathbf{p})$ is not satisfied, penetration occurs. Then $\mathbf{p}$ is projected to $\mathbf{p}'$:

$$\mathbf{p}' = \mathbf{p} - \nabla\phi(\mathbf{p})\phi(\mathbf{p}). \tag{6}$$

*3.3.3 Multi-Layerd Intersection Handling.* The clothing with $N$ layers of garments are aligned initially in canonical space and intersections may occur between any two layers. Fortunately, we have obtained the SDF $\phi$ for each clothed body in canonical space as described in Section 3.1, allowing us to leverage the SDF of a layer to correct penetrating vertices of other layer's mesh according to Equation (5) and Equation (6). Considering real-world scenarios, the innermost layer of clothing is usually closely fitted to the body, which is suitable to serve as the reference, hence penetration correction is performed from the outermost layer towards the inner layers. Since the goal is to eliminate penetration between garments, to avoid the influence of other parts of the clothed body, we apply a mask $\mathcal{R} \in \mathbb{R}$ to the SDF, which is determined by the bounding box $\mathcal{X} \in \mathbb{R}^{2\times3}$ of the corresponding garment extracted from this clothed body. If a query point is not in $\mathcal{X}$, we discard its collision constraint event it is penetrated. However, it is insufficient to rely solely on SDF to eliminate penetration. For a three-layer clothing, we first eliminate the penetration of the mesh of layer 3 based on the SDFs of layers 1 and 2. When handling the penetration of layer 2 based on the SDF of layer 1, penetration may occur again between layers 2 and 3. We discuss this further in Section 4.3.1. To address this problem, we devise the air constraint $C^{\text{air}}$ to keep the gap between two adjacent intersection-free layers (represented as $\mathcal{G}$ and $\hat{\mathcal{G}}$). Given $\mathbf{p} \in \mathcal{G}$, we have:

$$C^{\text{air}}(\mathbf{p}, \hat{\mathcal{G}}) = C^{\text{str}}(\mathbf{p}, \hat{\mathbf{q}})$$
$$\hat{\mathbf{q}} = \arg\min_{\mathbf{q} \in \hat{\mathcal{G}}} ||\mathbf{p} - \mathbf{q}||, \tag{7}$$

where $\mathbf{p}$ and $\hat{\mathbf{q}}$ are uniquely corresponding and pre-computed at the beginning of the simulation.

The complete procedure is outlined in Algorithm 1, where $N$ is the number of layers, $\mathcal{G}^i$ is the garment mesh of $i$-th layer, and $\phi^j$ is the clothed body SDF of $j$-th layer. Lines 2 to 7 correct the $i$-th layer, while lines 8 to 12 are implemented to ascertain that a gap is maintained between $(i + 1)$-th layer and $i$-th layer.

## 4 EXPERIMENTS

*Datasets. People Snapshot*[4] is a widely recognized dataset including monocular self-rotating videos[18, 23, 29, 45]. However, it only contains a few types of tight-fitting clothing, which is insufficient to support a comprehensive evaluation. Therefore, we

---

**Algorithm 1** Handling intersections from outside to inside.

```
1:  for i = N, N − 1, . . . , 2 do
2:      for all vertices p ∈ G^i do
3:          for j = 1, 2 . . . , i − 1 do
4:              solve collision R^j C^coll(p, φ^j)        ▷ Equation (5)
5:          end for
6:          solve stretch C^str(p, p̄)                    ▷ Equation (4)
7:      end for
8:      if i < N then
9:          for all vertices p ∈ G^{i+1} do
10:             solve air C^air(p, G^i)                    ▷ Equation (7)
11:         end for
12:     end if
13: end for
```

create *mini-IFG* that involves self-captured sequences (*mini-IFG-real*) and synthetic data (*mini-IFG-sim*) in a popular clothing design software, Style3D[1]. *mini-IFG* consists of 23 videos of 8 subjects with fashionable clothing. In each video, the actor rotates slowly in front of the camera while ensuring an uncovered garment as the target. Such measures allow for the evaluation of reconstructing both single-garment and multi-layered clothing.

*Baselines.* We compare with state-of-the-art (SOTA) works including video-based methods of SCARF[18] and REC-MV[45] and image-based of BCNet[24], SMPLicit[15], and ClothWild[39]. Since REC-MV doesn't release the model for detecting garment feature lines, we only reproduce *People Snapshot* for it.

*Metrics.* For quantitative comparison, we first align the estimated mesh to ground truth (synthetic data) by Iterative Closest Point (ICP) and then compute the Chamfer Distance (CD)[37] between them, where lower is better.

### 4.1 Single Garment Recontruction

Here, we aim to compare our approach with SOTA methods on the accuracy and generalization in single garment reconstruction. To avoid the impact of incomplete observations, we only reconstruct a single uncovered garment in each video and obtain the result of the first frame. For image-based methods[15, 24, 39], we input the first frame of the video. Similarly, for video-based methods[18, 45], we also compute the results of the first frame.

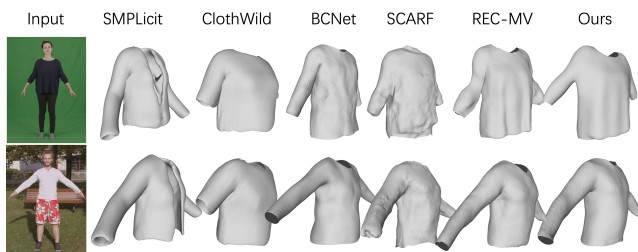

**Figure 4: Qualitative comparison on *People Snapshot*. Most methods can model such garments with simple topology.**

**Table 2: Quantitative results on synthetic sequences (*mini-IFG-sim*). We compare the Chamfer Distance (CD) between the ground-truth and reconstructed surfaces (in *cm*).**

| Method | Female-a | | Female-b | | | Male-a | | | Male-b | | |
|---|---|---|---|---|---|---|---|---|---|---|---|
| | Layer-1 | Layer-2 | Layer-1 | Layer-2 | Layer-3 | Layer-1 | Layer-2 | Layer-3 | Layer-1 | Layer-2 | Layer-3 |
| SMPLicit[15] | 2.9552 | 3.1213 | 1.9412 | 2.9713 | 2.5360 | 3.5042 | 4.0089 | 4.7301 | 1.7433 | 2.7373 | 3.4295 |
| ClothWild[39] | 2.5154 | 2.7793 | 3.1969 | 1.6661 | 4.0820 | 3.6386 | 4.2489 | 5.6636 | 2.4353 | 3.4968 | 4.9829 |
| BCNet[24] | 3.5378 | 1.9966 | 1.0470 | 1.0520 | 1.6689 | 3.6667 | 2.8670 | 3.4494 | **0.9508** | 2.5136 | 2.8745 |
| SCARF[18] | 2.3716 | 2.7509 | 4.2129 | 4.0528 | 3.5365 | 3.5762 | 4.4049 | 4.5656 | 2.4269 | 2.6494 | 2.9991 |
| Ours | **1.2689** | **1.9152** | **1.0215** | **1.0281** | **1.2331** | **1.5865** | **0.9943** | **1.6717** | 1.2363 | **1.0536** | **1.1297** |

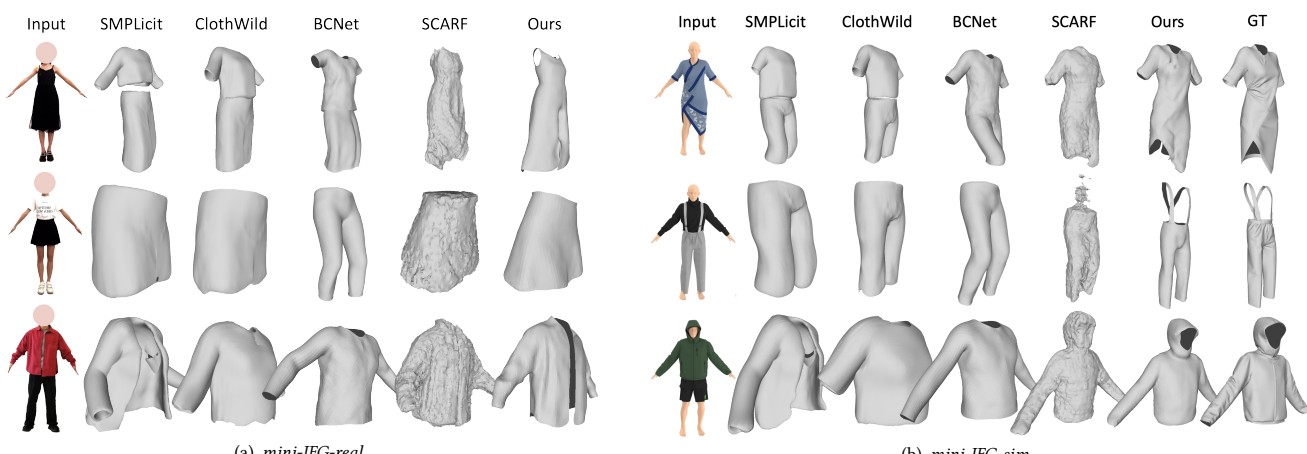

(a) *mini-IFG-real*

(b) *mini-IFG-sim*

**Figure 5: Qualitative comparison on *mini-IFG*. The goal is to reconstruct the uncovered garment in the image/video. SCARF[18] exhibits good generalization but low quality. Template-based methods[15, 24, 39] lack both detail and generalization. Our template-free approach demonstrates outstanding generalization, confidently handling extremely challenging garments.**

*4.1.1 Evaluation on Real-world Videos.* We show only two typical sequences from *People Snapshot* due to the limited variety of available garments. In Figure 4, most methods can handle such simple clothing. Among them, the template-based methods of REC-MV[45] and BCNet[24] behave close to ours. *mini-IFG-real* includes videos of humans wearing fashionable clothing. As shown in Figure 5(a), our method significantly outperforms others, thanks to the template-free extraction approach. Given the almost infinite variety of clothing styles, template-based methods[15, 24, 39, 45] struggle to be applicable in real life. Although SCARF[18] is also template-free, it suffers from poor geometric quality. In contrast, we achieve excellent generalization and high quality by effectively combining 2D segmentation with 3D implicit neural fields.

*4.1.2 Evaluation on Synthesis Videos.* Table 2 presents the quantitative results testing on *mini-IFG-sim*. We visually compare the reconstruction quality in Figure 5(b). For simple clothing, BCNet[24] and our method perform comparably. However, for challenging clothing, only our method can obtain correct results. SCARF[18] achieves shape similarity but exhibits significant noise, while others[15, 24, 39] deviate significantly from the ground truth. We demonstrate impressive results by reconstructing challenging clothing such as cheongsams and dungarees, distinctively.

*4.1.3 Dynamic Reconstruction.* As described in Section 3.2, clothed bodies in both the canonical space and posed space share a consistent geometric topology, which allows us to support dynamic reconstruction as well. We provide examples in Figure 6.

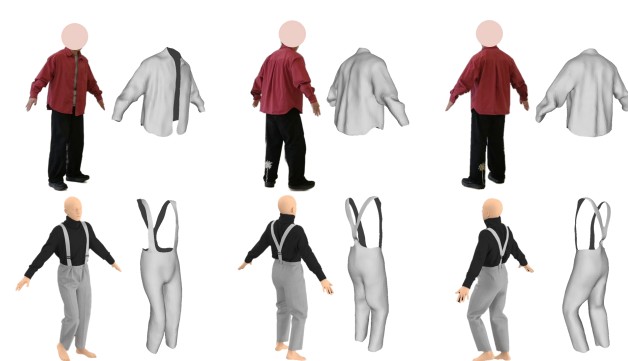

**Figure 6: Examples of dynamic reconstruction. For a single garment, since the mesh structure is shared across frames, we can accomplish dynamic capture.**

Figure 7: Qualitative comparison of multi-layered clothing reconstruction on *mini-IFG*. SCARF[18] treats clothing as a single entity. The template of BCNet[24] only supports a pair of tops and bottoms. SMPLicit[15] and ClothWild[39] can handle multi-layered clothing but lack detail. In contrast, we advocate for reconstructing multi-layered outfits from several videos. We are the only method that faithfully reconstructs multi-layered clothing while ensuring high quality and no penetration.

## 4.2 Multi-Layered Clothing Reconstruction

For each subject, our method inputs multiple videos, while baselines only input one video containing the outermost layer. This is because the baselines focus on the reconstruction from a single video but do not support the combination of clothing from multiple videos. Here, we aim to demonstrate the superiority of reconstructing multi-layered clothing based on multiple videos through comparison with baselines. Moreover, the robustness of penetration handling will also be validated.

Figure 7 provides qualitative comparisons. SCARF[18] exhibits the generality of the template-free method but regards clothing as a whole. The template in [24] is insufficient to meet fashion garments' requirements. Though layered clothing is available in SMPLicit[15] and ClothWild[39], the results lack detail and fail in the case of 4 layers. Fundamentally, since the mutual occlusion of clothing within a single video leads to incomplete observations, they can not obtain multi-layered clothing with great completeness. Therefore, additional videos are necessary, which hardly increases the usage difficulty. However, simply aligning the results from multiple videos inevitably causes penetration. We overcome this issue through a physics-aware module that robustly eliminates intersections, as

depicted in the right of Figure 7. For more detailed visualization, please refer to the supplementary material.

## 4.3 Ablation Study

*4.3.1 Constraints in Physics-Aware Module.* Here, we adequately illustrate the influence of constraints in the physics-aware module. First, we align the uncovered garments extracted from multiple videos in the canonical space (see Figure 8(a)). There are severe penetrations due to perceptual errors and overlapping deformations. Then, Figure 8(b) and Figure 8(c) depict the penetration correction strategy from the outer to the inner, as discussed in Section 3.3. Obviously, the intersection between layer 3 and layer 2 is alleviated in Figure 8(b) and the same occurs for layer 2 and layer 1 in Figure 8(c). For Figure 8(b) and Figure 8(c), we show constraints activated sequentially from top to bottom. For the top row of Figure 8(b), with only $C^{\mathrm{coll}}$, the penetration between layer 3 and layer 2 is noticeably improved but not completely eliminated (as seen at the cuff). This is because the simulation objects are discrete mesh vertices, and the sign distance values of vertices cannot reliably represent the penetration state of triangle faces. For the top row of Figure 8(c), some vertices of layer 2 are pushed out from the interior of layer 1 and then intersect with layer 3 again (at the hem of the top layer).

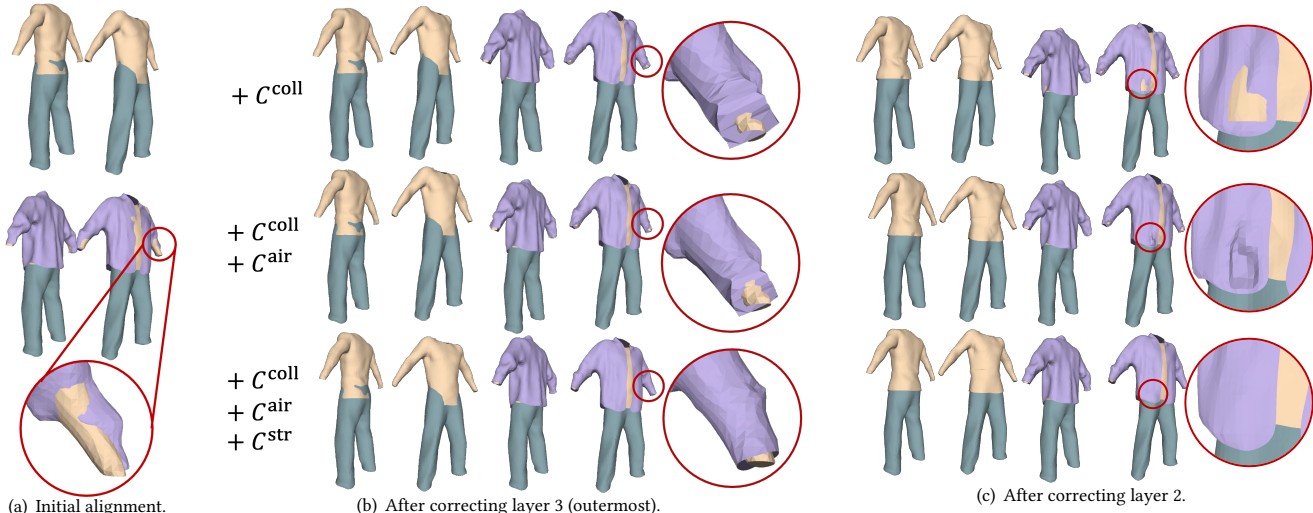

(a) Initial alignment.

(b) After correcting layer 3 (outermost).

(c) After correcting layer 2.

**Figure 8: Ablation study on the physics-aware module. In (a), there are penetrations of the initial alignment of three pieces of garments. In (b) and (c), we first correct the intersection of the outermost layer and follow with layer 2. From top to bottom of (b) and (c), different constraints are enabled in order. With $C^{coll}$, $C^{air}$, and $C^{srt}$, our proposed physics-aware module demonstrates outstandingly robust penetration handling capability.**

Fortunately, with $C^{air}$ to maintain gaps, the repeated inter-layer intersections are resolved (middle row of Figure 8(c)). However, the corrections brought by $C^{air}$ result in non-smoothness compared to areas where no penetration occurred around. Subsequently, after enabling $C^{str}$, non-rigid deformation eliminates this distortion (bottom row of Figure 8(c)). It is interesting that the penetration at the cuffs also disappears (bottom row of Figure 8(b)). This is attributed to our original pipeline in Algorithm 1. After solving $C^{coll}$, some vertices are projected to non-penetrating positions, accompanied by excessive stretching of edges. Then, $C^{str}$ pulls them closer by a certain distance. Since $C^{coll}$ and $C^{air}$ are solved once while $C^{str}$ is iteratively satisfied, they reach a balance as the simulation progresses: no penetration and no excessive deformation.

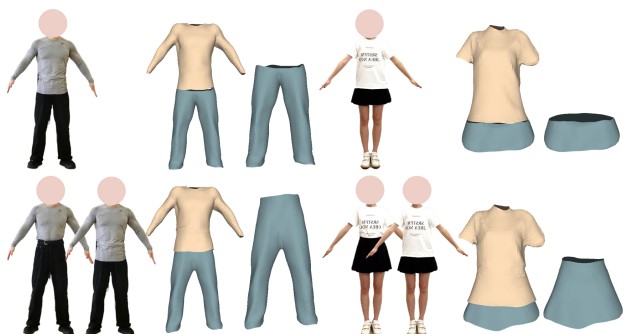

**Figure 9: With back-projection described in Section 3.2, we can also extract two garments from a single video (top row), but it is incomplete compared to the multiple videos way that we actually adopted (bottom row).**

*4.3.2 Reconstruction Two Garments from Single Video.* In Figure 9, we claim that IF-Garments can reconstruct two pieces of clothing from one video. However, due to occlusion, the lower lacks completeness. Instead, with multiple videos, all garments are complete.

## 5 LIMITATIONS

*Segmentation.* While our proposed template-free clothing extraction method achieves impressive generalization, the segmentation errors negatively impact mesh quality. With the rapid development of research on automated and semi-automated segmentation[25, 28, 32, 44], we believe this issue will be alleviated.

*Animation.* Currently, we are reconstructing multi-layered clothing in the canonical space. Since the deformation fields corresponding to each layer of clothing are independent, penetration occurs again when garments are mapped to the posed space. So IF-Garments is unsuitable for direct usage in animation. Recently, some research related to clothing simulation has contributed to multi-layered clothing animation[9, 10]. It is possible to achieve penetration-free posed meshes by leveraging their results.

## 6 CONCLUSION

We have presented IF-Garments, a novel framework for multi-layered intersection-free clothing reconstruction from monocular videos. Our core innovation lies in ingeniously combining neural SDFs with back-projection and physics simulation to accomplish both remarkable generalization of clothing extraction and robustness of handling intersections. Sufficient experiments thoroughly demonstrate the superiority and effectiveness of our method. Due to convenience and high quality, we believe that IF-Garments can benefit downstream multimedia tasks such as human performance capture, personalized avatar modeling, and virtual try-ons.

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
