# OpenReview forum: "IF-Garments: Reconstructing Your Intersection-Free Multi-Layered Garments from Monocular Videos"
_acmmm.org/ACMMM/2024/Conference — MM2024 Poster_

### Official Review · Reviewer_F5DG · 2024-04-28

**Rating:** 4
**Confidence:** 2

**Summary:**

The paper presents IF-Garments, a novel solution for reconstructing multi-layered garments without intersections. By leveraging semantic segmentation to guide the learning of neural Signed Distance Fields (SDFs), the method enables accurate reconstruction of layered clothing from monocular videos. Additionally, it employs computer graphics techniques to effectively address the issue of overlapping garments, ensuring physically plausible reconstructions.

**Strengths:**

1. The paper proposes a seemingly practical solution for reconstructing multi-layered garments and alleviating occlusion issues, which, while having a notably engineering-centric flavor, displays technical feasibility.
2. The proposed method, though adopting a rudimentary yet effective approach of using videos featuring subjects in varied clothing as input for template-free garment reconstruction, proves technically sound and holds instructive value for related research.
3. Complementing its well-structured content, the paper boasts high-quality illustrations that significantly augment the overall readability.

**Limitations:**

1. The article seems to rely solely on existing Neural SDF techniques for reconstruction, employing multiple iterations to obtain separate layers of clothing, which may not constitute a significant advancement in reconstruction technology.
2. The physically-based method for dealing with overlap presented herein is grounded in traditional graphics approaches, involving the manipulation of mesh vertices, an approach that does not introduce evidently new technological innovations.
Therefore, these points cumulatively cast doubt on the paper's technological originality.

**Suitability:**

3

---

### Official Review · Reviewer_cJdu · 2024-05-24

**Rating:** 4
**Confidence:** 3

**Summary:**

The proposed method reconstructs clothed bodies from monocular videos and utilizes image segmentation labels to obtain multiple garments. To eliminate inter-layer penetration, the method proposes a physics-aware module to fine-tune the garments from the outer to the inner layer. The experiments validate the effectiveness of the proposed method.

**Strengths:**

1. The proposed physics-aware module can effectively resolve the multi-layered clothing intersection issue.
2. This paper propose a novel dataset that involves self-captured sequences with clothed humans.
3. This paper is well-written and easy to follow.

**Limitations:**

1. When there is occlusion for multi-layer garments in the 2D segmentation labels, and since these are self-rotating sequences, this occlusion cannot be resolved through multi-views.  how are the occluded 3D vertices assigned segmentation labels? It seems that the proposed method can merely apply in scenarios with multiple videos containing multiple unoccluded garments.

2. The proposed approach can be time-consuming, which may hinder its adoption in real-time applications or situations that require rapid reconstruction.

**Suitability:**

3

---

### Official Review · Reviewer_wgvX · 2024-05-24

**Rating:** 3
**Confidence:** 3

**Summary:**

This paper aims to reconstruct multi-layered clothing from monocular videos, enabling cost-effective and convenient digitalization of garments. This work holds significant practical implications for multimedia applications such as gaming, metaverse, and virtual reality.

**Strengths:**

+ A template-free approach with great generalization for extracting a complete garment.

+ A physics-aware module ensuring multi-layered clothing intersection-free with excellent robustness and quality

+ A small dataset containing 23 self-rotating videos of actors wearing fashionable garments.

**Limitations:**

-1. The proposed method seems to be more favorable towards reconstructing male models. Please analyze this phenomenon.

-2. Can the proposed method show results for different poses?

-3. The ablation experiments lack data analysis.

-4. There is a lack of comparative experiments between this method and other Intersection-free methods ([2], [51]).

-5. There is a lack of quantitative and qualitative experiments between this method and baseline methods for 2023.

If the author can effectively address my concerns, I'm willing to raise my rating.

**Suitability:**

2

---

### Official Review · Reviewer_2WRV · 2024-05-24

**Rating:** 4
**Confidence:** 3

**Summary:**

This paper presents a novel framework (IF-Garments) for multi-layered intersection-free clothing reconstruction from monocular videos. It combines neural SDFs, back-projection, and physical simulation, increasing the generalization and robustness of multiple clothing extraction and integration. In addition, a novel dataset is created, which provides more monocular self-rotating videos with both real and simulated scenarios.

**Strengths:**

1. As mentioned in this paper, their proposed method is the first attempt to reconstruct multi-layered clothing using multiple monocular videos.
2. This method combines physical simulation and introduces multiple physics constraints, which achieves more reasonable and accurate modeling of multiple clothing.
3. This paper proposes a new monocular self-rotating video dataset that involves diversity and challenging clothing, providing more data access for subsequent related tasks.

**Limitations:**

1. I would like to know the potential of this work for other downstream tasks, such as whether it can facilitate clothing replacement. If possible, providing some related examples would be more convincing and meaningful.
2. I noticed that when handling the multi-layered intersection, the representations of clothing on different layers are not the same (mesh or SDF), I want to know what the distinctions are between them and why are they configured in this manner.
3. Based on the qualitative results provided in the paper, it is apparent that the current work struggles to effectively model certain clothing details, such as the hem or wrinkles of the skirt.
4. Please confirm that the title of Section 3.3.3 is spelled correctly.

**Suitability:**

3

---

### Meta-Review · Area_Chair_KUdX · 2024-07-07

**Recommendation:** Accept (Poster)
**Confidence:** 5

**Metareview:**

All reviewers have agreed to accept this paper. The authors need to address all the reviewers' comments in the revised manuscript.